# Setup uncertainties and appropriate setup margins in the head-tilted supine position of whole-brain radiotherapy (WBRT)

**Jae Won Park[1,2], Ji Woon Yea[1,2], Jaehyeon Park[1,2], Se An Oh** [1,2] *

**1** Department of Radiation Oncology, Yeungnam University Medical Center, Daegu, Korea, **2** Department of Radiation Oncology, Yeungnam University College of Medicine, Daegu, Korea

* sean.oh5235@gmail.com

## Abstract

Various applications of head-tilting techniques in whole-brain radiotherapy (WBRT) have been introduced. However, a study on the setup uncertainties and margins in head-tilting techniques has not been reported. This study evaluated the setup uncertainties and determined the appropriate planning target volume (PTV) margins for patients treated in the head-tilted supine (ht-SP) and conventional supine position (c-SP) in WBRT. Thirty patients who received WBRT at our institution between October 2020 and May 2021 in the c-SP and ht-SP were investigated. The DUON head mask (60124, Orfit Industries, Wijnegem, Belgium) was used in the c-SP, and a thermoplastic U-Frame Mask (R420U, Klarity Medical & Equipment Co. Ltd., Lan Yu, China) was used in the ht-SP. Daily setup verification using planning computed tomography (CT) and cone-beam CT was corrected for translational (lateral, longitudinal, and vertical) and rotational (yaw) errors. In the c-SP, the means of systematic errors were -0.80, 0.79, and 0.37 mm and random errors were 0.27, 0.54, and 0.39 mm in the lateral, longitudinal, and vertical translational dimensions, respectively. Whereas, for the ht-SP, the means of systematic errors were -0.07, 0.73, and -0.63 mm, and random errors were 0.75, 1.39, 1.02 mm in the lateral, longitudinal, and vertical translational dimensions, respectively. The PTV margins were calculated using Stroom et al.'s [2Σ+0.7σ] and van Herk et al.'s recipe [2.5Σ+0.7σ]. Appropriate PTV margins with van Herk et al.'s recipe in WBRT were <2 mm and 1.5˚ in the c-SP and <3 mm and 2˚ in the ht-SP in the translational and rotational directions, respectively. Although the head tilt in the supine position requires more margin, it can be applied as a sufficiently stable and effective position in radiotherapy.

## Introduction

Radiotherapy for patients with brain tumours are based on various factors, such as the size, location, and cell type of the primary tumour. Whole-brain radiotherapy (WBRT) is a commonly used effective technique to relieve neurological symptoms, improve the quality of life, and prolong survival [1–5].

In our previous study [6], we have demonstrated that head-tilting techniques can be effectively used to reduce radiation exposure of normal tissues, such as the hippocampus, lens, and

**Data Availability Statement:** All relevant data are within the manuscript.

**Funding:** "This work was supported by the 2021 Yeungnam University Research Grant. The funders had no role in the study design, data collection and

analysis, decision to publish, or preparation of the manuscript".

**Competing interests:** The authors have declared that no competing interests exist.

parotid gland, using volumetric arc therapy (VMAT). Moreover, the radiation conformity and dose distribution in tumours simultaneously improved.

Recently, various studies using a head-tilting baseplate in radiotherapy have been published. Miura et al. [7] compared the potential of the use of tomotherapy in the dose distribution improvement of the planning target volume (PTV) and reduction of exposure time of normal organs, such as the hippocampus and lens, with and without the head-tilting baseplate in hippocampal-sparing WBRT. Tilted hippocampal-sparing WBRT reportedly reduced the radiation exposure time by more than 10% of normal organs such as the hippocampus and lens.

Shimizu et al. [8] used the four-field box technique in WBRT to examine the best head-tilt angle that could reduce the parotid gland dose while maintaining a safe level of lens dose. Since the parotid dose is inversely proportional to the lens dose, it was concluded that the orbitomeatal plane angle required to reduce the maximum lens dose to less than 10 Gy and minimise the parotid gland dose was 14˚.

Lin et al. [9] investigated the various head flexion angles in hippocampal-sparing WBRT in VMAT. This study demonstrated that at least 15˚ should be implemented in clinical practice, and for better dose coverage and uniformity of whole-brain PTV and dose reduction in critical organs, a head angle of $\geq 25$˚ is recommended.

Therefore, the application of the head tilt in WBRT has several advantages. However, data on the setup uncertainties in head-tilting techniques in WBRT and the clinical target volume (CTV) to PTV setup margins has not been reported. Although the head-tilt technique has several advantages in radiotherapy, if the setup uncertainties are excessively large, the PTV margin becomes large and the dose delivered to the normal organs, such as the hippocampus, parotid, and lens, increases; thus, the existing advantages of the head-tilt technique may be negated. Therefore, it is necessary to study the exact setup uncertainties and PTV margins for the head-tilt technique compared to that of the existing supine setup.

Hence, this study evaluated the setup uncertainties and determined the appropriate PTV margins for patients treated in the conventional supine position (c-SP) and head-tilted supine position (ht-SP) using a head-tilting baseplate in WBRT.

## Materials and methods

### Study overview

This retrospective data analysis of the 30 patients who underwent WBRT enrolled in this study was approved by the Institutional Review Board of the Yeungnam University Medical Center (YUMC 2021-07-033). Informed consent was specially waived under the approval of the institutional review board, given that patient anonymity was ensured.

### Patient selection

All patients who received WBRT between October 2020 and May 2021 at our institution were included in the study. Thirty patients who received radiotherapy in the c-SPs and ht-SP fixed with a thermoplastic mask were investigated. The patient radiotherapy techniques and treatment schedules included in this study are described in Table 1. Of the 30 WBRT patients, 15 were treated in the c-SP and 15 in the ht-SP. Patients treated in the c-SP had one patient who received 25 Gy in 10 fractions, and 14 patients who received 30 Gy in 10 fractions for a total fraction number of 150. Meanwhile, among the patients treated in the ht-SP, one patient received 25 Gy in 10 fractions, and 14 patients received 30 Gy in 10 fractions for a total fraction number of 150.

**Table 1. Characteristics of the patients and treatment included in this study.**

|  | c-SP | Ht-SP |
|---|---|---|
| **Number of patients** | N = 15 | N = 15 |
| **Number of fractions** | n = 150 | n = 150 |
| **Median age (Range)** | 59(35–83) | 62(37–86) |
| **Sex (%)** |  |  |
| **Female** | 9(60) | 7(46.7) |
| **Male** | 6(40) | 8(53.3) |
| **Radiotherapy techniques (%)** |  |  |
| **3DRT** | 2(13) | 15(100) |
| **VMAT** | 13(87) | 0(0) |
| **Fraction schemes (%)** |  |  |
| **25 Gy in 10 fractions** | 1(7) | 1(7) |
| **30 Gy in 10 fractions** | 14(93) | 14(93) |

3D = three-dimensional radiotherapy; VMAT = volumetric-modulated arc therapy.

## Immobilisation and CT simulation

Fig 1 shows the setup position of the patient in the supine and the head-tilted supine position with a thermal mask before radiotherapy in the radiation treatment room. A thermal plastic mask was used in all the patients to minimise the inter- and intra-fractional variations of radiotherapy. In the traditional c-SP, the DUON head mask (60124, Orfit Industries, Wijnegem, Belgium) was fixed at 2.4-mm mask thickness. For the ht-SP, a tilting acrylic supine baseplate (MedTec, USA) was used to elevate the patient's head to up to 40° according to our institution's protocol, and a thermoplastic U-Frame Mask (R420U, Klarity Medical & Equipment (GZ) Co. Ltd., Lan Yu, China) with 2.4-mm mask thickness was used. All the computed tomography (CT) simulation images were obtained using a Brilliance Big Bore CT simulator (Philips Inc., Cleveland, OH) with a thickness of 2.5–5 mm.

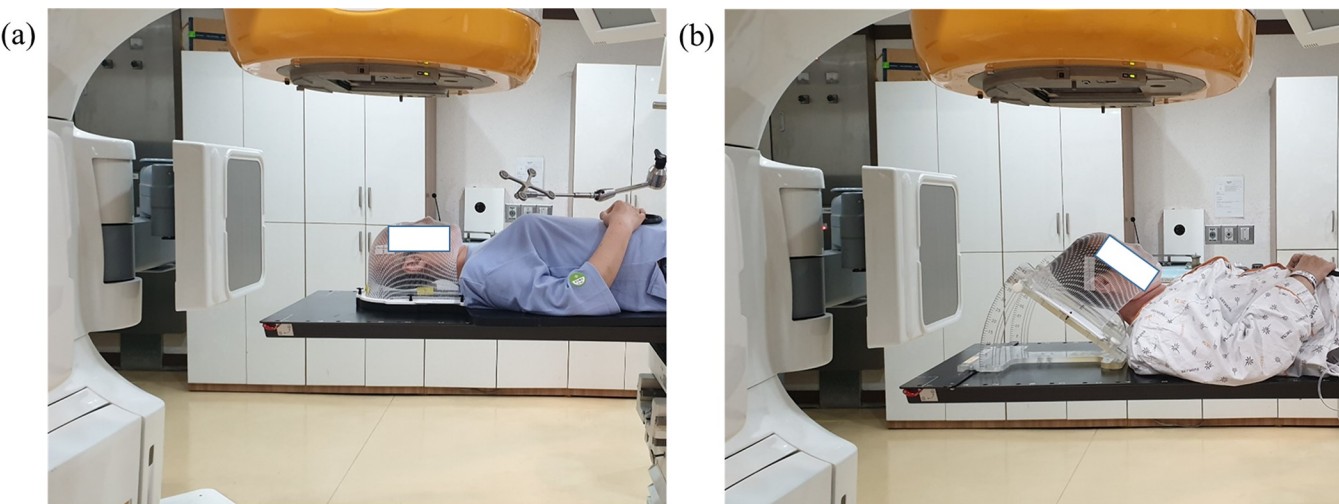

(a)          (b)

**Fig 1. Patient setup position for radiotherapy.** (A) supine position, (B) head-tilted supine position.

## Treatment planning and delivery techniques

We performed delineation of CT and MRI images of T2-weighted and gadolinium contrast-enhanced T1-weighted sequences using a rigid fused MRI-CT image set. The CTV of the whole brain was defined as the parenchyma and the spinal cord up to the lower level of the atlas. In the three-dimensional radiotherapy (3DRT) and volumetric-modulated arc therapy (VMAT) techniques, the PTV was created with an extension of 5 mm in all directions from the CTV. An anisotropic analytic algorithm (AAA Varian Eclipse TPS, version 15.6.05) was used for all the radiation treatment plans as reported in previous studies [6,10]. The Novalis-Tx (Varian Medical System, CA, USA and BrainLAB, Feldkirchen, Germany) linear accelerator system with a high-definition multi-leaf collimator and 6 degrees of freedom (DOF) robotic couch (BrainLAB, Feldkirchen, Germany) was used. However, cone-beam CT (CBCT) images in the translational (lateral, longitudinal, and vertical) and rotational (yaw) directions applicable to 4DOF were used. Photon energy of 6 MV was used in all radiation treatment planning. 3DRT used a beam arrangement in 0˚, 90˚, 180˚, and 270˚ directions, and the VMAT used two coplanar full-arc beams with clockwise (CW) and counter clockwise (CCW) gantry rotation. The collimator of the VMAT was rotated around 290˚ for CW and 25˚ for CCW to minimize the tongue-and-groove effect.

## Image registration and setup protocol

The image registration using planning CT and CBCT in the c-SP and the ht-SP is shown in Figs 2 and 3, respectively. Daily setup verification images were obtained for all 30 patients who received WBRT using CBCT (Varian Medical Systems, Palo Alto, CA, USA) before treatment. The X-ray tube voltage and current used for CBCT imaging were 100 kV and 80 mA, respectively. Figs 2A and 3A show the planning CT images in transverse, frontal, and sagittal directions, and Figs 2B and 3B were obtained with CBCT in pre-treatment. Figs 2C and 3C show the registration images between the planning CT and CBCT images in the translational and rotational directions. To match the planning CT and CBCT during image registration, an experienced therapist adjusted the window level appropriately. Also, in the image registration, a clip box was used to include PTV. Image registration between the planning CT and CBCT image was performed using bony anatomy auto-matching. There is no institutionally accepted shift tolerance, and all radiation treatment patients were treated with radiation by correcting the translational and rotational directions based on the differences in the planning CT and pre-treatment CBCT. The automatic corrections of the couch on the setup differences in the translational and rotational directions were recorded.

## Analysis of the setup uncertainties between the planning CT and CBCT

A one-sample Kolmogorov–Smirnov test was performed to verify the normal distribution of all the recovered setup corrections. A non-parametric Mann–Whitney U-test was performed to compare the systematic and random setup errors of the two independent groups between the c-SP and the ht-SP. The mean values of the random and systematic errors are denoted as mean ±standard deviation. All the statistical tests were performed using the SPSS statistical software version 22 (SPSS Inc., Chicago, IL, USA), and a $p$-value $<0.05$ was considered statistically significant.

Van Herk et al. introduced a method to analyse the random ($\sigma$) and systematic errors ($\Sigma$) using setup correction values for setup verification and have been applied in several papers [11–13]. During the image registration process, the differences in four directions, translational (lateral, longitudinal, and vertical) and rotational (yaw) directions, were analysed between the planning CT and CBCT images. Several studies [11–20] have already been conducted on the

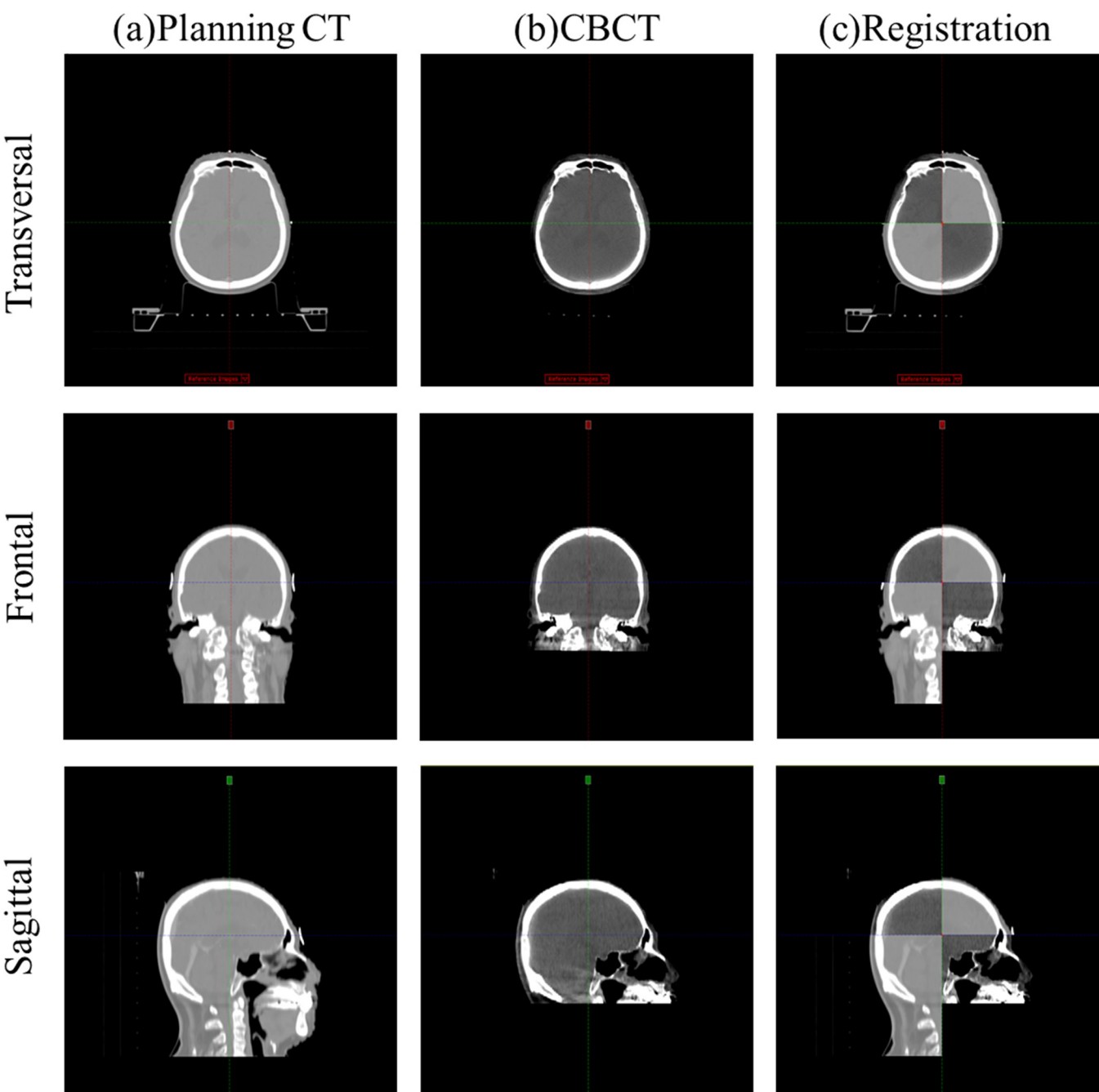

**Fig 2. Image registration using the planning computed tomography (CT) and cone beam CT (CBCT) in the c-SP.** (A) Planning CT, (B) CBCT, and (C) registration image.

appropriate PTV margin recipe through expansion from CTV. In this study, the PTV margin was calculated using the methods of Stroom et al. [14] and Van Herk et al. [15].

Stroom et al. [14] assumed a 95% dose, on an average of 99% of the CTV tested in a practical setting. However, this study has only been demonstrated in prostate, cervix, and lung cancers. Conversely, Van Herk et al. [15] assumed that the minimum dose for the CTV was 95% for 90% of the patients using the analytical solution for perfect conformation. However, Van

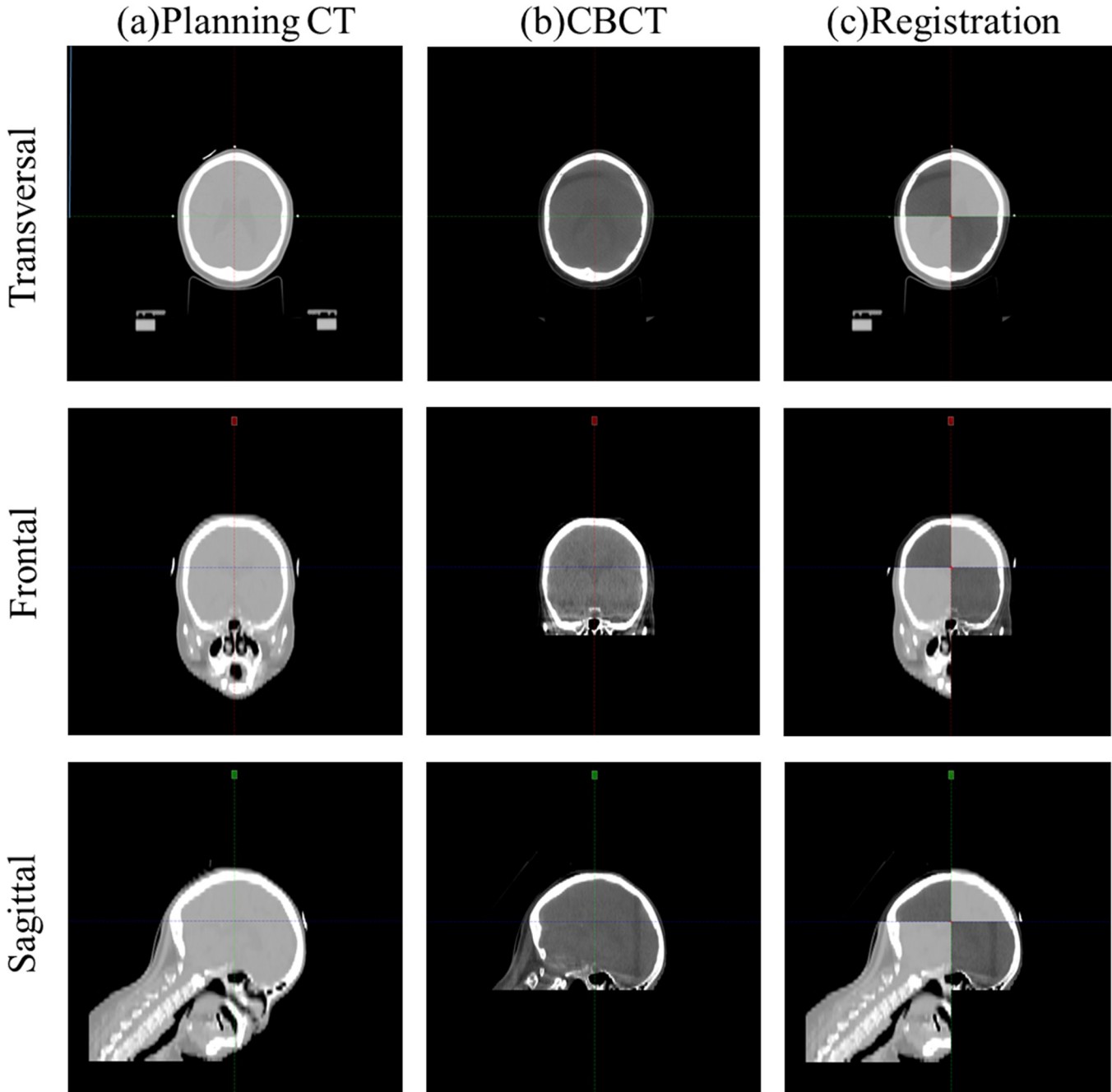

**Fig 3. Image registration using the planning computed tomography (CT) and cone beam CT (CBCT) in the ht-SP.** (A) Planning CT, (B) CBCT, and (C) registration image.

Herk et al. reported that because the margin excludes rotational errors and shape deviations, this recipe must be considered as a lower limit for safe radiotherapy.

$$\text{Stroom et al.}'\text{s recipe} = 2\Sigma + 0.7\sigma$$

$$\text{Van Herk et al.}'\text{s recipe} = 2.5\Sigma + 0.7\sigma$$

where $\Sigma$ is the systematic errors, and $\sigma$ is the random errors.

We also compared the 3D vector values between the c-SP and ht-SP. The 3D vector values can be calculated as follows:

$$3D \ vector = \sqrt{x^2 + y^2 + z^2}$$

where x, y, and z are the errors in the lateral, longitudinal, and vertical directions, respectively.

## Results

Figs 4 and 5 show the histograms and normal distribution curves of the setup errors in translation (lateral, longitudinal, and vertical) and rotation (yaw) in the c-SP and ht-SPs, respectively.

Table 2 shows the systematic ($\Sigma$) and random errors ($\sigma$) in the translational and rotational directions for the c-SPs and ht-SPs. In the c-SP, the mean values of the systematic error in the lateral, longitudinal, vertical, and yaw directions were -0.80±0.47 mm, 0.79±0.34 mm, 0.37±0.54 mm, and 0.17˚±0.56, respectively. The mean values of the random errors were 0.27±0.24 mm, 0.54±0.31 mm, 0.39±0.20 mm, and 0.35˚±0.17, respectively. The mean values of the systematic error in the lateral, longitudinal, vertical, and yaw directions in the ht-SP were -0.07±1.10 mm, 0.73±0.97 mm, -0.63±0.83 mm, and -0.01˚±0.69, respectively. The mean values of the random errors were 0.75±0.16 mm, 1.39±0.42 mm, 1.02±0.28 mm, and 0.57˚±0.26, respectively.

Fig 6 shows the results of the box plot and Mann–Whitney U-test in translation and rotation for the setup errors between the c-SP and ht-SPs. All the data were not normally

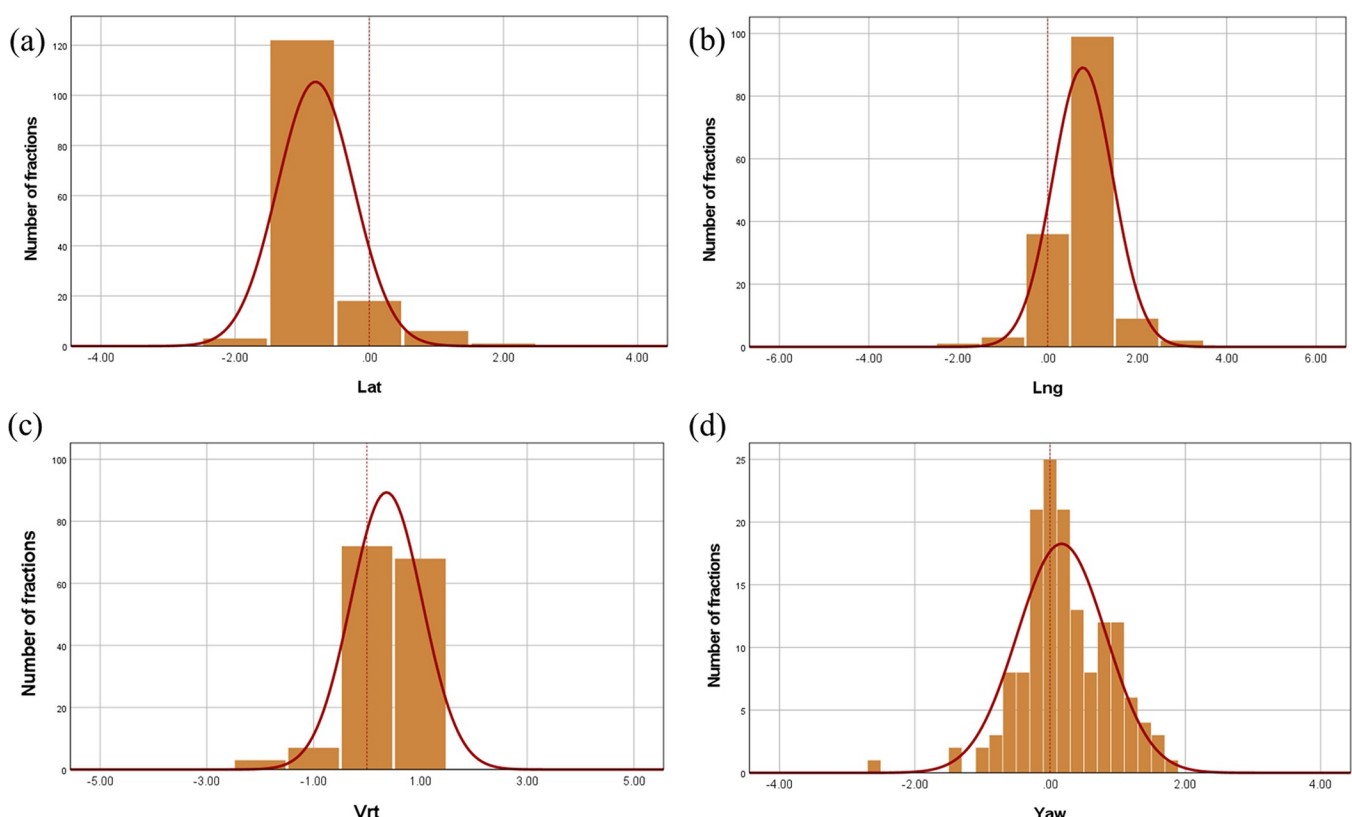

**Fig 4. Histograms and normal distribution curves of the setup errors in the translation and rotation for the supine position (c-SP).** Setup errors in the (A) lateral, (B) longitudinal, (C) vertical, and (D) yaw directions.

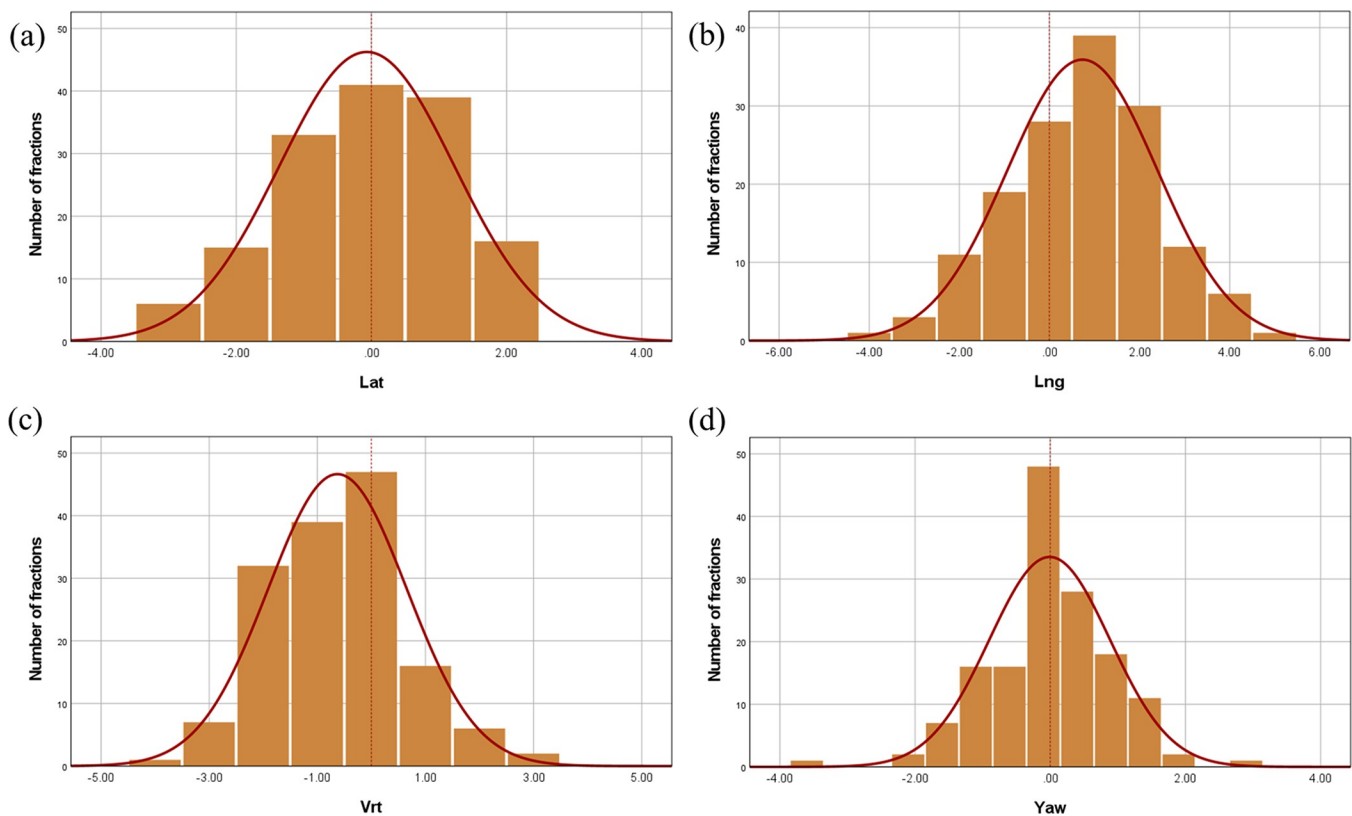

**Fig 5. Histograms and normal distribution curves of the setup errors in the translation and rotation for the head-tilted supine position (ht-SP).** Setup errors in the (A) lateral, (B) longitudinal, (C) vertical, and (D) yaw directions.

distributed in any direction. There was a statistically significant difference in the lateral and vertical directions, and there was no significant difference in the longitudinal (*p*-value = 0.941) and yaw (*p*-value = 0.109) directions.

The results of the box plot and Mann–Whitney U-test for the 3D vector between the c-SP and ht-SP are shown in Fig 7. The *p*-value was <0.001, indicating a significant difference.

Table 3 shows the CTV to PTV margin calculated by Stroom et al.'s and van Herk et al.'s margin recipe for systematic (Σ) and random errors (σ). In the c-SP, the CTV to PTV margin

**Table 2. Systematic errors (Σ) and random errors (σ) in the translational (lateral [x-axis], longitudinal [z-axis], and vertical [y-axis]) and rotational (yaw [y-axis]) directions.**

| Setup errors | c-SP | | | | Ht-SP | | | |
|---|---|---|---|---|---|---|---|---|
| | Systematic errors (Σ) | | Random error (σ) | | Systematic errors (Σ) | | Random errors (σ) | |
| | Mean (mm) | S.D | Mean (mm) | S.D | Mean (mm) | S.D | Mean (mm) | S.D |
| **Translational** | | | | | | | | |
| Lateral (x-axis) (mm) | -0.80 | 0.47 | 0.27 | 0.24 | -0.07 | 1.10 | 0.75 | 0.16 |
| Longitudinal(z-axis) (mm) | 0.79 | 0.34 | 0.54 | 0.31 | 0.73 | 0.97 | 1.39 | 0.42 |
| Vertical (y-axis) (mm) | 0.37 | 0.54 | 0.39 | 0.20 | -0.63 | 0.83 | 1.02 | 0.28 |
| **Rotational** | | | | | | | | |
| Yaw (y-axis) (°) | 0.17 | 0.56 | 0.35 | 0.17 | -0.01 | 0.69 | 0.57 | 0.26 |

SD = standard deviation.

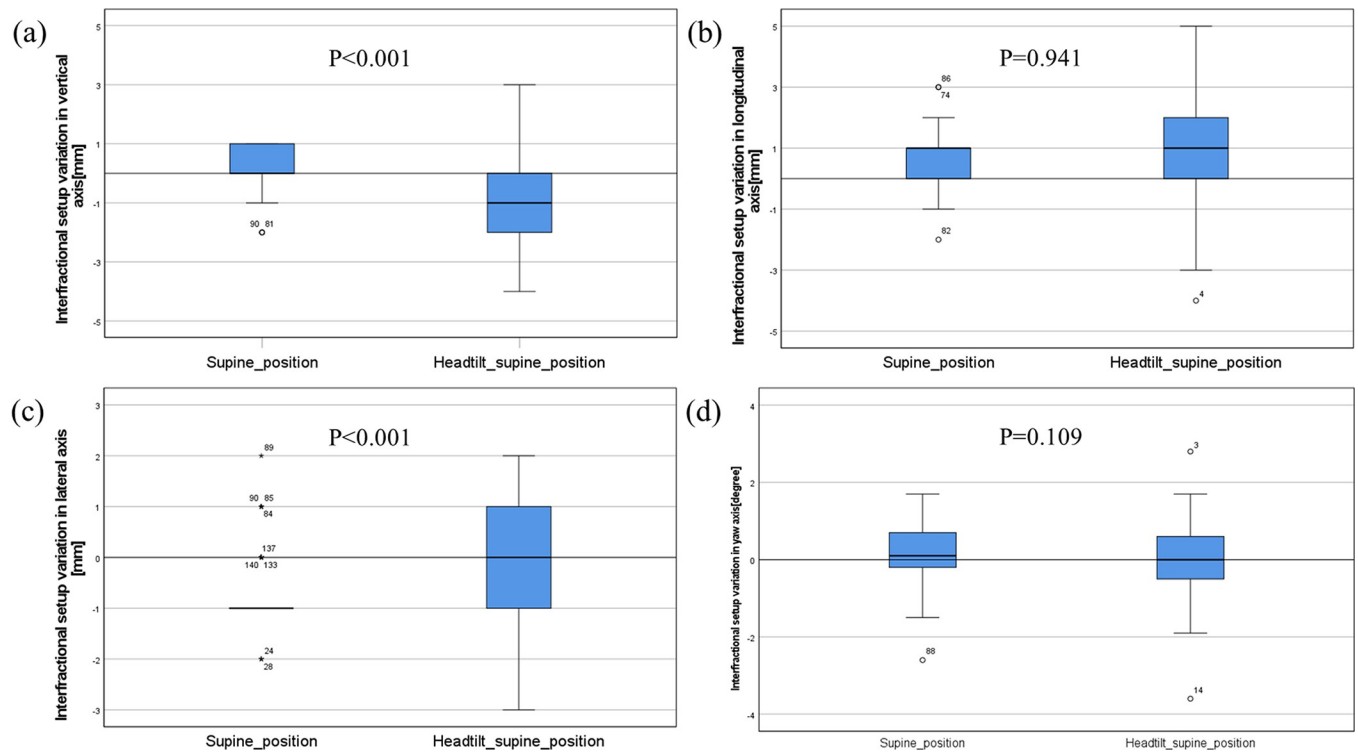

**Fig 6. Boxplot and *p*-value of the Mann–Whitney U-test in the translation and rotation be-tween the c-SP and ht-SP.** Setup errors in the (A) lateral, (B) longitudinal, (C) vertical, and (D) yaw directions.

was 1.1 mm, 0.9 mm, 1.2 mm, and 1.2˚ with Stroom et al.'s recipe for lateral, longitudinal, vertical, and yaw directions, and 1.3 mm, 1.1 mm, 1.5 mm, and 1.5˚ with van Herk et al.'s recipe, respectively. The CTV to PTV margin in the ht-SP was 2.3 mm, 2.2 mm, 1.9 mm, and 1.6˚ with Stroom et al's recipe, and 2.9 mm, 2.7 mm, 2.3 mm, and 1.9˚ with van Herk et al's recipe, respectively. In the CTV to PTV margin, the ht-SP was larger than c-SP in all directions.

## Discussion

In this study, the setup uncertainties and appropriate PTV margins were determined by analysing the setup errors of CBCT images of patients before radiotherapy in the c-SP (N = 15) and ht-SP using a head-tilting baseplate (N = 15) in WBRT.

According to the results of our recent study [6], the conformity and homogeneity indices of the target were improved upon using the head-tilting baseplate VMAT in hippocampal-sparing WBRT, and the mean dose to normal tissues, such as the hippocampus, parotid, and right and left lenses, were significantly reduced. However, one of the limitations of the previous study was that the dose delivered to the lens was relatively higher in the supine VMAT than in the head-tilted VMAT because the treatment plan was optimised to maximise target coverage and spare the hippocampus. Therefore, it would have been difficult to preserve the hippocampus in the supine VMAT plan because the lens constraint was a high priority.

In another study, Miura et al. reported [7] that a tilted hippocampus-sparing WBRT with tomotherapy while sparing healthy organs, including the hippocampus and lens, could reduce the treatment time by more than 10%. However, their study had the limitation of a small sample size of five; therefore, a larger cohort study should be conducted. Furthermore, because

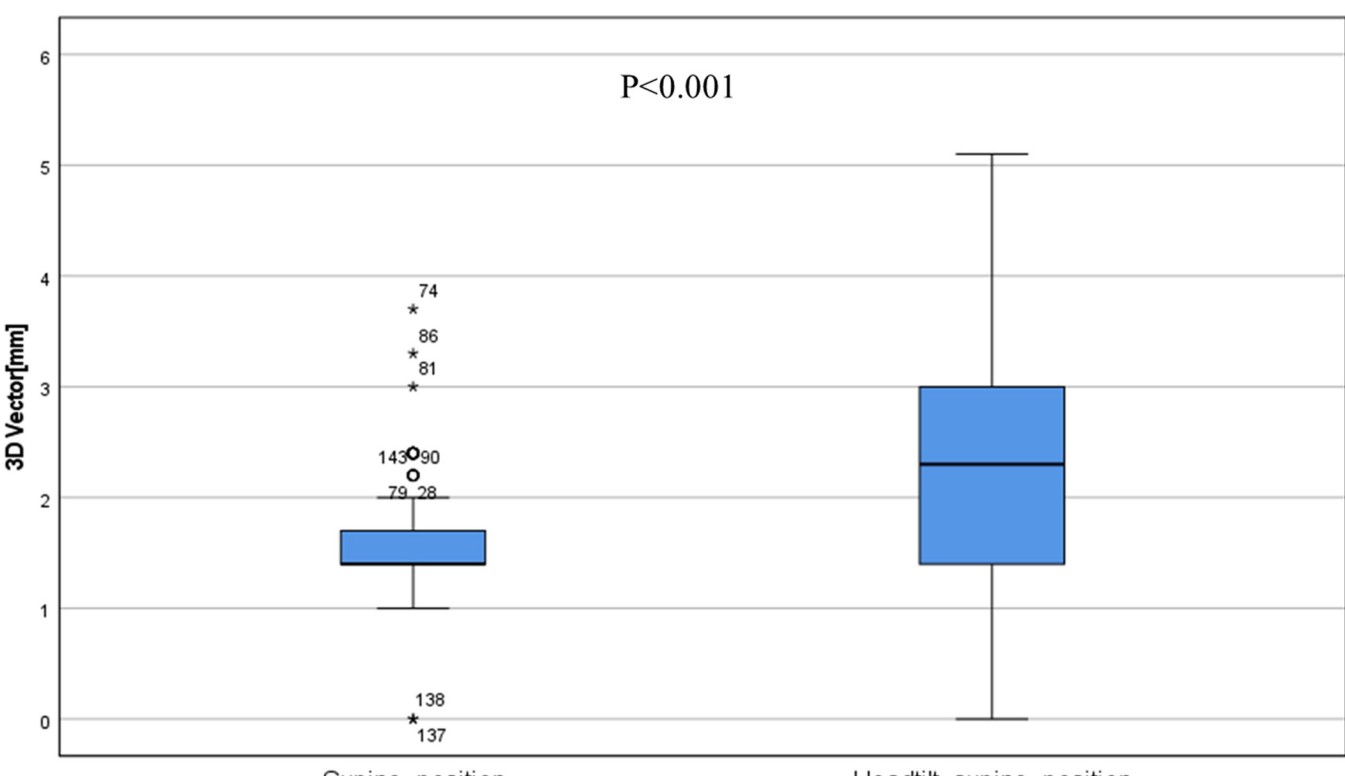

**Fig 7. Boxplot and *p*-value of the Mann–Whitney U-test in the three-dimensional radiotherapy (3D) vector between the c-SP and ht-SPs.**

they used two pairs of CT images on the same day, they were able to evaluate two CT images and construct a useful deformable image registration algorithm.

Several studies have reported the advantages of the ht-SP in radiotherapy [6–9]; however, no studies have reported on the uncertainties and PTV margins for this setup. The PTV should be extended from the CTV with an appropriate margin. However, less expansion of the CTV could lead to an increase in the uncertainty of the radiation dose to the PTV resulting in

**Table 3. Clinical target volume (CTV) to planning target volume (PTV) margin calculations calculated by Stroom et al.'s [14] and van Herk et al.'s [15] margin recipe for systematic ($\Sigma$) and random errors ($\sigma$).**

| Studies | Setup position | Imaging modalities | Recipe | PTV margin | | | |
|---|---|---|---|---|---|---|---|
| | | | | Translational | | | Rotational |
| | | | | Lateral (mm) | Longitudinal (mm) | Vertical (mm) | Yaw (°) |
| **Oh et al. [18]** | Supine | CBCT | Van Herk et al | 3.73 | 3.45 | 3.24 | |
| **Zhou et al [19]** | Supine | MVCT | Stroom et al | 4.8 | 5.0 | 1.5 | |
| **Oh et al. [12]** | Supine | ExacTrac | Stroom et al | 1.7 | 3.5 | 2.3 | 1.9 |
| **Present Study** | Supine | CBCT | Stroom et al | 1.1 | 0.9 | 1.2 | 1.2 |
| | | | Van Herk et al | 1.3 | 1.1 | 1.5 | 1.5 |
| | Head tilt supine | | Stroom et al | 2.3 | 2.2 | 1.9 | 1.6 |
| | | | Van Herk et al | 2.9 | 2.7 | 2.3 | 1.9 |

*Used formula: Stroom et al.'s recipe[14] = $2\Sigma+0.7\sigma$, Van Herk et al.'s recipe [15] = $2.5\Sigma+0.7\sigma$.

CBCT = cone-beam computed tomography; MVCT = megavoltage computed tomography.

undesired radiation treatment outcomes. However, if the margin of the CTV is expanded too large, a sufficient radiation dose can be delivered to the CTV; however, at the same time, normal tissues may be exposed [21]. Therefore, radiation treatment positions, such as the c-SP and ht-SP, which can affect setup uncertainty in radiotherapy, should be investigated for appropriate margin calculations.

Oh et al. [18] analysed the setup uncertainties in 21 (438 fractions) cases of brain tumours using daily CBCT. The patient was immobilised in the c-SP using a thermoplastic fixation mask, and the registration procedure between the acquired CBCT and the planning CT image was performed according to the bony anatomy. They used van Herk et al.'s recipe [15] to calculate the PTV margin, which was 3.73 mm, 3.45 mm, and 3.24 mm in the lateral, longitudinal, and vertical directions, respectively.

Similar to their study, the PTV margin in the present study using the image tool of CBCT in the c-SP was 1.3 mm, 1.1 mm, 1.5 mm, and 1.5˚ with van Herk et al.'s recipe [15], and 1.1 mm, 0.9 mm, 1.2 mm, and 1.2˚, in the lateral, longitudinal, vertical, and yaw directions with Stroom et al.'s recipe [14], respectively. Whereas the PTV margins in the ht-SP were 2.9 mm, 2.7 mm, 2.3 mm and 1.9˚ with van Herk et al.'s recipe., and 2.3 mm, 2.2 mm, 1.9 mm, and 1.6˚ with Stroom et al.'s recipe, respectively.

Therefore, when the ht-SP was compared to the c-SP in WBRT, it was observed that the PTV should be further expanded by 1.2 mm, 1.3 mm, 0.7 mm, and 0.3˚ in the lateral, longitudinal, vertical, and yaw directions, respectively, as per Stroom et al.'s recipe. Moreover, according to van Herk et al.'s recipe, the PTV should be expanded further by 1.5, 1.6, 0.8, and 0.4˚, respectively. In our study, although a little more margin was required in the ht-SP than in the c-SP. Therefore, further research is needed to reduce the setup margin in WBRT treatment in the c-SP and ht-SP in the future.

Appropriate CTV to PTV margin analysis through systematic and random error analysis can be evaluated using various image tools, such as CBCT [18], megavoltage CT [19], and ExacTrac [12], and the results can be compared as shown in Table 3. The factors causing the differences in the setup uncertainties in the translational and rotational directions according to the setup position in WBRT should be evaluated. As shown in Table 2, it seems that the mask was unable to fix the position of the patient and the baseplate in the ht-SP compared to in the c-SP. Therefore, random error is considered larger in the ht-SP. Although the head-tilt in the supine position requires a little more margin, it can be utilized as a safe and effective position in radiotherapy. Therefore, additional studies are needed on the advantages of sparing organs at risk and the disadvantages of a little more margin in the head-tilt supine position.

This study has some limitations. First, the patients who received radiotherapy in the c-SP and ht-SP did not use the same mask. This is because the treatment position and mask were determined according to our institution's setup protocol since the setup accuracy varied based on the thermoplastic mask [22,23]. Second, our study analysed images using rigid image registration in the image analysis between the planning CT and pre-treatment verification CBCT. The use of deformable image registration could yield varying results [24–27]. Third, since this study was calculated considering only inter-fractional variation, the PTV margin could be different if intra-fractional variation was included [28–30]. Fourth, since we used only 40˚ to elevate the patient's head according to our institution's protocol, this study could not demonstrate the effect of different angles on setup uncertainties.

## Conclusions

This study analysed the setup error of 15 patients treated in the c-SP and 15 in the ht-SP using a head-tilting baseplate in WBRT. In the c-SP, the means of systematic errors were -0.80, 0.79,

and 0.37 mm and random errors were 0.27, 0.54, and 0.39 mm in the lateral, longitudinal, and vertical translational dimensions, respectively. Whereas, for the ht-SP, the means of systematic errors were -0.07, 0.73, and -0.63 mm, and random errors were 0.75, 1.39, 1.02 mm in the lateral, longitudinal, and vertical translational dimensions.

Appropriate PTV margins with van Herk et al.'s recipe in WBRT were <2 mm and 1.5˚ in the c-SP, and <3 mm and 2˚ in the ht-SP in the translational and rotational directions, respectively. Although the head tilt in the supine position requires a little more margin, the head-tilt can be applied as a safe and effective position in radiotherapy.

## Author Contributions

**Conceptualization:** Jae Won Park, Ji Woon Yea, Jaehyeon Park, Se An Oh.

**Funding acquisition:** Se An Oh.

**Investigation:** Jae Won Park.

**Methodology:** Jae Won Park, Ji Woon Yea, Jaehyeon Park.

**Project administration:** Se An Oh.

**Resources:** Ji Woon Yea, Jaehyeon Park, Se An Oh.

**Supervision:** Se An Oh.

**Validation:** Jae Won Park, Ji Woon Yea, Jaehyeon Park, Se An Oh.

**Writing – original draft:** Jae Won Park.

**Writing – review & editing:** Jae Won Park, Ji Woon Yea, Jaehyeon Park, Se An Oh.

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
