## [Decision Letter · Decision Letter 0]

18 Apr 2022

PONE-D-22-03374Setup uncertainties and appropriate setup margins in the head-tilted supine position of whole-brain radiotherapy (WBRT)PLOS ONE

Dear Dr. Oh,

Thank you for submitting your manuscript to PLOS ONE. After careful consideration, we feel that it has merit but does not fully meet PLOS ONE’s publication criteria as it currently stands. Therefore, we invite you to submit a revised version of the manuscript that addresses the points raised during the review process.

We look forward to receiving your revised manuscript.

Kind regards,

Ngie Min Ung

Academic Editor

PLOS ONE

Journal Requirements:

“This work was supported by the 2021 Yeungnam University Research Grant.”

“This work was supported by the 2021 Yeungnam University Research Grant.”

“This work was supported by the 2021 Yeungnam University Research Grant.”

Additional Editor Comments (if provided):

Please address the comments from all reviewers, particularly comments from Reviewer 2.

Reviewers' comments:

Reviewer's Responses to Questions

**Comments to the Author**

1. Is the manuscript technically sound, and do the data support the conclusions?

Reviewer #1: Yes

Reviewer #2: No

Reviewer #3: Yes

2. Has the statistical analysis been performed appropriately and rigorously? 

Reviewer #1: Yes

Reviewer #2: No

Reviewer #3: Yes

3. Have the authors made all data underlying the findings in their manuscript fully available?

Reviewer #1: Yes

Reviewer #2: No

Reviewer #3: Yes

4. Is the manuscript presented in an intelligible fashion and written in standard English?

Reviewer #1: Yes

Reviewer #2: No

Reviewer #3: No

5. Review Comments to the Author

Reviewer #1: Whole brain radiotherapy (WBRT) is one of the most effective radiotherapy-based approaches to treat patient with brain metastasis. Due to the complex anatomical structures within the treatment fields, more conformal treatment is essential to provide good local control rates with less late toxicity. There are some published literatures reported the use of a head-tilting baseplate in WBRT and its advantages in comparison with conventional supine position setup. Typical CTV-PTV margins are 3-5 mm (Ann Barrett, Jane Dobbs et al, 2009) to eliminate the systematic and random errors. To date, there is lack of publications discussing the setup uncertainties and CTV-PTV margins with regards to the head-tilting baseplate setup technique. Therefore, this study is relevance and able to provide general interest to the readers of the journal.

This study performed image registration on planning CT and cone beam CT in order to determine the setup uncertainties; and adapted the methods proposed by Stroom et al. [2Σ+0.7σ] and van Herk et al. [2.5Σ+0.7σ] to estimate the CTV-PTV margins. The authors found that the head-tilting baseplate setup technique was sufficiently stable and able to provide effective position in WBRT, while proposing the CTV-PTV margins of <3 mm. In general, I found that this paper is well structured. Nevertheless, there is lack of detailed elaborations for some of the descriptions or important points. With that in view, I suggest that a minor revision should be considered and the authors should address the comments as detailed below:

Major comments:

(1) The objectives of the study were to evaluate the setup uncertainties and determine the appropriate CTV-PTV margins for patients treated with or without a head-tilting baseplate. It is recommended that the abstract and conclusion sections should also include the summaries of setup uncertainty.

(2) The authors have chosen Stroom et al. and van Herk et al. recipes in estimating CTV-PTV margins. Please mention the rational and limitation (if any) of these selections. For instance, van Herk et al.’s (2000) method was excluding rotational errors and shape deviations, and considered as a lower limit for safe radiotherapy. Stroom et al.’s (1999) method was initially applied to prostate, cervix, and lung cancer case.

(3) Page 14, line 260: The authors have mentioned that the CTV-PTV margins that applied in the clinic were 5 mm. However, the study revealed that the margin should expand further compared to the conventional supine position. Please elaborate further in the discussion section how these findings have changed the current clinical practice if any?

Minor comments:

(4) Table 1: Please check the table settings (merge/unmerge) of each column titles to avoid confusion.

(5) Page 6, lines 101: Please check the angle of beam directions.

(6) Page 8, line 131 and page 9, line 155: The symbols () and () denote different parameter in this manuscript, i.e. random error, systematic error and its standard deviation. Please make necessary amendment in this manuscript to avoid confusion.

(7) Page 8, line 137: Suggest to move the sentence “All the data were not normally distributed in any direction” to Result section.

(8) Page 9, line 158 & line 170-181: Please rephrase the paragraphs as some of them were repeatedly mentioned in the text as well as figures 4&5 or Table 2.

(9) Table 2’s caption: Please check whether pitch [x-axis] and roll [z-axis] were included in your data set.

(10) Table 3: Data for “vertical” should be under “translational” column.

(11) Table 3: Please check the value for “vertical-Stroom recipe- Head tilt supine”, 1.9 instead of 1.8.

(12) Page 15, line 272: Repetition of conclusion “In conclusion, …effective position in radiotherapy.”

(13) Figure 4-7: the font size of the axis labels was too small.

(14) Reference: Please ensure the reference formatting is in line with the journal requirements, for instance:

a. Reference 1: Use the abbreviation “Surg. Neurol. Int.”

b. Reference 13: Use the abbreviation “Int. J. Radiat. Oncol. Biol. Phys.”

Reviewer #2: This work focused on PTV margin calculated from daily setup errors in two different patients' treatment setup during whole brain radiotherapy, which may contribute to the scientific knowledge in radiotherapy field. However, the inability of the authors to present proper and accurate analysis and discussion, hence producing low quality of scientific manuscript. Each sections needs to be re-write in scientific article style in order for higher appreciation of the data presented in this study. This manuscripts suffers from the lack review of related literatures in addition to the inability of the authors to perform varieties of analysis with the available data and also the inability to calculate the margin according to the well-known Van Herk or Stroom formulas. It is hope that the author would revise and re-analyze all data pertaining to this work in order to achieve the objective of the study. It would be my pleasure to review again the revised manuscript, should the author performed a proper and correct data analysis as this study do has significant contribution to radiotherapy field.

Reviewer #3: The study investigates the uncertainties of whole brain radiotherapy in a head-tilt position. The manuscript has described the work in sufficient details. The following can be considered to improve the manuscript

1. The write-up has too many short paragraphs of single/two sentences that can be combined with other paragraphs.

2. Abstract: The motivation of using head-tilting technique can be briefly highlighted in the abstract

3. Line 23: Please rephrase the sentence. The sentence is describing the image correction not the radiotherapy delivery. “Radiotherapy was performed by correcting the translational (lateral, longitudinal, and vertical) and 24 rotational (yaw) errors based on the differences in computed tomography (CT) and cone-beam CT.”

4. Suggest to compare results obtained with recently published margin, e.g., http://doi.org/10.1002/acm2.13291

5. Line 272, In discussion, although the head tilt gives larger margin, perhaps authors can comment on how the benefit of sparing OAR, can justify the use of head-tilt? This can be included in the conclusion briefly.

6. Perhaps authors can discuss how the head-tilt technique can be improved, to reduce the uncertainties.

6. PLOS authors have the option to publish the peer review history of their article (what does this mean?). If published, this will include your full peer review and any attached files.

Reviewer #1: No

Reviewer #2: No

Reviewer #3: **Yes: **HM Zin

---

## [Author Response · Author response to Decision Letter 0]

30 May 2022

we attached point by point responses to all reviewers and editor.

---

## [Decision Letter · Decision Letter 1]

23 Jun 2022

Setup uncertainties and appropriate setup margins in the head-tilted supine position of whole-brain radiotherapy (WBRT)

PONE-D-22-03374R1

Dear Dr. Se An Oh,

We’re pleased to inform you that your manuscript has been judged scientifically suitable for publication and will be formally accepted for publication once it meets all outstanding technical requirements.

Kind regards,

Ngie Min Ung

Academic Editor

PLOS ONE

Additional Editor Comments (optional):

Please address the remaining minor corrections in the current draft.

Reviewers' comments:

Reviewer's Responses to Questions

**Comments to the Author**

1. If the authors have adequately addressed your comments raised in a previous round of review and you feel that this manuscript is now acceptable for publication, you may indicate that here to bypass the “Comments to the Author” section, enter your conflict of interest statement in the “Confidential to Editor” section, and submit your "Accept" recommendation.

Reviewer #1: All comments have been addressed

Reviewer #2: All comments have been addressed

Reviewer #3: All comments have been addressed

2. Is the manuscript technically sound, and do the data support the conclusions?

Reviewer #1: Yes

Reviewer #2: Yes

Reviewer #3: Yes

3. Has the statistical analysis been performed appropriately and rigorously? 

Reviewer #1: Yes

Reviewer #2: Yes

Reviewer #3: Yes

4. Have the authors made all data underlying the findings in their manuscript fully available?

Reviewer #1: Yes

Reviewer #2: Yes

Reviewer #3: Yes

5. Is the manuscript presented in an intelligible fashion and written in standard English?

Reviewer #1: Yes

Reviewer #2: Yes

Reviewer #3: Yes

6. Review Comments to the Author

Reviewer #1: Thanks to the authors who had revised the manuscript accordingly. However, further improvement should be made to address several minor corrections. With that in view, I suggest that a minor revision should be considered and the authors should address the comments as detailed below:

Minor corrections:

1. Page 3, line 41 & Page 17, line 312: Please tally the c-SP random errors results with data presented in Table 2.

2. Page 7, line 128: Suggest to revise the manufacturer’s details “The Novalis-Tx (Varian Medical System, CA, USA) linear accelerator system with a high-definition multi-leaf collimator and 6 degrees of freedom (DOF) robotic couch (BrainLAB, Feldkirchen, Germany) was used.”

3. Page 8, line 141: mA

4. Figure 4: Please enlarge the font size of the labels in the figure.

5. Table 3: The caption of the table didn’t reflect its contents which includes the comparison data with other studies. Please revise.

6. Page 15, line 282: Please double check the value for vertical which supposed to be 0.8 mm.

7. Page 15, line 284-288 & page 295, line 295-298: Please rephrase and consolidate the sentences to avoid repetition.

8. Acknowledgement: Please include acknowledgements wherever appropriate.

Reviewer #2: The revised version has strengthened the manuscript. I strongly recommend it for publication with minor correction (as suggested in the attached document) to enhanced the quality of published manuscript.

Reviewer #3: The paper has improved and addressed concerns from the reviewers. The pape can be accepted for publication.

7. PLOS authors have the option to publish the peer review history of their article (what does this mean?). If published, this will include your full peer review and any attached files.

Reviewer #1: No

Reviewer #2: No

Reviewer #3: **Yes: **Hafiz Zin

---

## [Editor Report · Acceptance letter]

27 Jul 2022

PONE-D-22-03374R1 

Setup uncertainties and appropriate setup margins in the head-tilted supine position of whole-brain radiotherapy (WBRT) 

Dear Dr. Oh:

I'm pleased to inform you that your manuscript has been deemed suitable for publication in PLOS ONE. Congratulations! Your manuscript is now with our production department. 

Kind regards, 

on behalf of

Dr. Ngie Min Ung 

Academic Editor

PLOS ONE